# Apomixis Technology: Separating the Wheat from the Chaff

**DOI:** 10.3390/genes11040411

**Published:** 2020-04-10

**Authors:** Diego Hojsgaard

**Affiliations:** Department of Systematics, Biodiversity and Evolution of Plants, Albrecht-von-Haller Institute for Plant Sciences, Georg-August-University of Göttingen, Untere Karspüle 2, D-37073-1 Göttingen, Germany; Diego.Hojsgaard@biologie.uni-goettingen.de

**Keywords:** apomeiosis, clonal seeds, endosperm, heterosis capture, molecular breeding, parthenogenesis

## Abstract

Projections indicate that current plant breeding approaches will be unable to incorporate the global crop yields needed to deliver global food security. Apomixis is a disruptive innovation by which a plant produces clonal seeds capturing heterosis and gene combinations of elite phenotypes. Introducing apomixis into hybrid cultivars is a game-changing development in the current plant breeding paradigm that will accelerate the generation of high-yield cultivars. However, apomixis is a developmentally complex and genetically multifaceted trait. The central problem behind current constraints to apomixis breeding is that the genomic configuration and molecular mechanism that initiate apomixis and guide the formation of a clonal seed are still unknown. Today, not a single explanation about the origin of apomixis offer full empirical coverage, and synthesizing apomixis by manipulating individual genes has failed or produced little success. Overall evidence suggests apomixis arise from a still unknown single event molecular mechanism with multigenic effects. Disentangling the genomic basis and complex genetics behind the emergence of apomixis in plants will require the use of novel experimental approaches benefiting from Next Generation Sequencing technologies and targeting not only reproductive genes, but also the epigenetic and genomic configurations associated with reproductive phenotypes in homoploid sexual and apomictic carriers. A comprehensive picture of most regulatory changes guiding apomixis emergence will be central for successfully installing apomixis into the target species by exploiting genetic modification techniques.

## 1. Introduction

With a world population expected to reach 9.7 billion by 2050 [1] and increasing food demand, plant breeders are expected to create more resilient crops to overcome constraints on cereal production imposed by population growth, climate change and environmental degradation [2].

The use of novel tools in biotechnology and the advances in the characterization of genes and genomes is playing a central role speeding up plant breeding efforts to increase yields and grain quality. However, the creation of a new crop variety takes between 7–20 years to bring to the market, costs from hundreds of thousands to millions of euros, and cannot capture all beneficial gene interactions [3,4,5]. Selected traits in high-yield hybrid cultivars segregate in the offspring during the (normal) mechanism of (sexual) reproduction and seed formation, and new hybrid seeds must be generated every year from selected parental lines to keep superior plant phenotypes. Projections indicate that the current breeding approaches will be unable to produce the increase in global crop yields needed to address global food security [6]. As long as the development of new hybrid varieties relies entirely in the exploitation of sexuality, breeding programs are inevitably costly and time-demanding. 

Apomixis is a disruptive innovation, an alternative to sex that can speed up the time and reduce the cost needed to create a new cultivar. By skipping key steps of sexuality (Figure 1), apomictic plants can produce clonal seeds, and hence, capture heterosis and gene combinations of elite phenotypes transgenerationally [7,8]. The potential of breeding apomixis into hybrid cultivars has been known for a long time [5] and scientists had been looking for ways to introgress, induce, or mimic apomixis in sexual crops with the central concept that pollen of apomicts could transmit dominant apomixis genes in crosses with sexual plants or that apomixis might be induced from altering the sexual development. Harnessing apomixis would accelerate breeding programs and the ability of plant breeders to fix and propagate genetic heterozygosity and the associated hybrid vigor, reducing the time required to produce new varieties as well as the costs associated with seed-production [5,7,8]. The introduction of apomixis into hybrid cultivars is a game-changing innovation in the current plant breeding paradigm. Apomixis breeding can lead to rapid mass production of new elite cultivars better adapted to local environments across the world, reduce the problems connected to monocultures, and contribute to crop resilience by meeting food production goals without restrictions from biotic and abiotic stresses, climate change, or environmental degradation.

Whereas the majority of flowering plants produce seeds following the events of sexual reproduction (Figure 1a), around 400 species (or ca. 0.1% of all angiosperm species) belonging to 293 genera [9,10] had evolved the singular reproductive mechanism called apomixis. The fact that (1) apomixis has recurrently arisen across time from sexual progenitors [11], (2) apomictic species naturally occur in diverse plant families throughout the angiosperm phylogeny [9,10], and (3) a few of those apomictic species are minor crops (such as some forage grasses, fruits, and guayule) [5,12] suggest that apomixis can be introduced and/or engineered into major cereal and legume crops. Thus far, attempts to introgress apomixis from wild relatives into important crop species such as maize, wheat, and pearl millet have failed mainly due to interspecific and ploidy barriers [13,14,15].

The search of the functional control of apomixis has been a goal of crop scientists for the past 70–100 years and has involved a plethora of different approaches with little or no success [16,17]. Identification of particular genes in sexual model plants associated with apomixis-like features date back to before the Bellagio apomixis declaration [18], and yet, limited progress has been done. In the last years, identification of new genes linked to apomixis-like phenotypes or apomixis components had opened new perspectives but had also found new constraints. Apomixis must coordinate molecular and developmental interactions between one sporophytic and two gametophytic tissues and it is certainly not a single gene trait (see details below), which adds hurdles to apomixis breeding [8,19]. Even if simultaneously modifying several key genes produces plants mimicking apomixis phenotypes, relevant knowledge about their genetic background and possible molecular interactions and responses to regulatory signals in developmental cascades during gametogenesis and embryo and endosperm developments will still be missing. This is likely the reason why characterized apomixis-like mutants often display low penetrance and paltry quality phenotypes (see discussion below and in [20]). In the case of apomixis technology, the central problem behind constraints to apomixis breeding is that the genomic background and molecular mechanisms that initiate apomixis and guide the formation of a functional clonal seed are still poorly understood. Thus, about the road to the creation of sexual plants genetically modified to produce seeds carrying non-recombinant embryos also implies resolving the current blindness about the molecular basis behind natural apomixis. Understanding what genetic mechanism and molecular changes elicits apomixis emergence in natural species will not only benefit current efforts toward synthetic apomixis but is also a central prerequisite to harness its potential use in breeding along with an effective (penetrant) expression in sexual, domesticated crop plants.

## 2. Developmental Features of Apomixis 

Apomictic plants can skip sex and produce seeds carrying clonal embryos. By doing so, ovules of apomictic plants modify key steps of cell specification, female meiosis, gamete fusion, and embryo and endosperm development. Two main types of apomixis had been described: sporophytic and gametophytic apomixis. In sporophytic apomixis (also called adventitious embryony), an embryo is directly formed from a somatic cell (e.g., integumentary or nucellar) in the ovule, which can occupy or attach to the meiotic embryo sac, thus rising seeds with poly-embryos [21,22]. Fertilization of the meiotic embryo sac is needed to produce a functional seed and for the growth of these embryos until maturity [22]. In gametophytic apomixis (from here onwards “apomixis”), the embryo is formed via an unreduced female gametophyte that engenders a mono-embryonic seed in most cases after fertilization of the endosperm. Plants showing gametophytic apomixis modify three components of the standard sexual program of seed formation. First, meiosis is prevented or skipped to form unreduced female gametophytes. The circumvention of meiosis (or apomeiosis) can happen 1) by differentiation of a nucellar cell into a megaspore-like cell independent of the megaspore mother cell (MMC) progression (apospory, Figure 1b) or 2) at the MMC via a failure in chromosome pairing (failing of the formation of the synaptonemal complex) and subsequent conversion of the first meiotic division into a mitosis-like division (diplospory, Figure 1c). In both cases, the unreduced megaspore or megaspore-like cell undergoes gametogenesis and generates an embryo sac [21,23]. The second key step is the lack of egg cell fertilization. In apomeiotic embryo sacs, the egg cell initiates embryogenesis by parthenogenesis, i.e., without fertilization [21]. The third crucial step to produce a functional clonal seed is the development of the endosperm (a nutritious tissue that regulate cellular differentiation and embryonic organogenesis [24]). In most apomictic plants, the endosperm usually develops after fertilization of the unreduced central cell (pseudogamy), which is insensitive or less deterrent to paternal ploidy misbalances than in meiotic embryo sacs [25,26]. Thus, the endosperm primary cell in apomicts tolerates shifts of parental genome dosages (from 2maternal:0paternal contributions in embryo sacs with autonomous endosperm development, and 1m:1p up until 8m:1p in pseudogamous embryo sacs), other than the 2m:1p genomic ratio required for endosperm development during sexual seed formation [27,28].

Furthermore, apomixis is almost exclusively associated with polyploidy in plants, often forming agamic complexes in which diploids are persistently sexual and polyploids arising from diploids are persistently apomictic [29,30]. In these complexes, co-occurring diploids show a tendency to produce low proportions of apomeiotic female gametophytes [31,32,33], and yet, functional apomixis is not stable except for dihaploids in *Erigeron* [34] and diploid cytotypes in *Boechera* [35,36]. In addition, apomictic plants are facultative, meaning that sexual reproduction is not completely eliminated [37]. Facultative plants keep a low level of sexual seed formation [38,39,40,41].

Thus, apomixis is not only a rare biological phenomenon but a developmentally complex trait.

## 3. Genetic and Genomic Features of Apomixis

The molecular genetics behind apomixis is still unsolved. The inheritance of apomixis has a multifaceted nature. Apomixis inheritance can be explained by Mendelian genetics, but every conceivable complication for genetic analysis seems to accumulate in apomicts [15]. The use of different molecular mapping techniques (comparative mapping, linkage disequilibrium mapping, and deletion mapping) and Sanger sequencing methods helped researchers to uncover details about the complex molecular background associated with apomixis as well as to identify sequences and genes linked to the trait. Studies carried out both in aposporous and diplosporous species (e.g., *Brachiaria*, [42,43]; *Cenchrus*/*Pennisetum*, [44,45]; *Erigeron*, [46]; *Hieracium*, [47]); *Hypericum* [48]; *Paspalum*, [49,50,51,52]; *Ranunculus* [53]; *Taraxacum* [54]; *Trypsacum*, [55]), indicates that apomixis behaves as a dominant trait over sexuality, with apomictic components co-segregating, often inherited as a simplex phenotype and exhibiting segregation distortion. Genetic factors linked to apomixis components had been localized into large chromosomal regions of up to 50 Mb showing suppression of recombination [50,56] and hemizygous linkage groups (i.e., markers do not show hybridization signals in sexuals [56,57,58]). Apomixis loci of monocots and dicots allocate chromosome structural changes (mainly inversions and translocations; [50,58,59]) and share microsynteny among different apomictic and sexual species [60], a feature that may relay—at least in grasses—upon recombinogenic (euchromatic) or nonrecombinogenic (heterochromatic) regions [61]. Mechanisms operating to reduce recombination and retain linkage disequilibrium could be ascribed to allele sequence divergence, chromosomal rearrangements, and chromatin remodeling to enhance the formation of heterochromatin. BAC clones carrying markers linked to apomixis show presence of numerous rearrangements due to insertions or deletions of transposable elements interrupting gene sequences [57,62,63,64]. Increased allelic sequence divergence, retrotransposon activity and mutational degradation [65,66] are expected evolutionary consequences of chromosomal regions subjected to repression of meiotic recombination. The evolution of the heteromorphic chromosomal region in organisms with sexual chromosomes is a good example [67,68]. These features reduce the efficiency of map-based cloning strategies and the genetic dissection of apomixis.

Attempts to identify components of apomixis by transcriptional profiling of reproductive organs using Differential Display PCR (DD-PCR), SuperSAGE (serial analysis of gene expression), and high-throughput sequencing on microdissected ovules revealed 1) differentially expressed genes in reproductive tissues of apomictic and sexual relatives from different plant systems (e.g., *Pennisetum*, [69,70]; *Brachiaria*, [71,72]; *Panicum*, [73,74]; *Poa*, [75], *Eragrostis*, [76,77,78]; *Paspalum*, [42,64,79,80,81]; *Hieracium*, [82,83]; *Hypericum* [84,85]) and 2) an overall shift in gene regulation at the MMC stage and a global heterochronic gene expression between the sexual and apomeiotic ovules (e.g., *Boechera*, [86,87,88,89]; *Paspalum* [79,90,91]; *Pennisetum*, [92]; *Hieracium*, [83,93,94]; *Hypericum* [85,95]; *Ranunculus* [96,97]). Such wide-ranging de-regulation on gene expression levels between sexual and apomictic ovules affect genes encoding varied biological functions (GO classes) and regulatory pathways, including key genes of the sexual pathway, RNA-directed DNA methylation and transcription regulation, hormonal signaling, and cell cycle control (see details in the next section). All these changes in gene expression, modulation of gene networks, cellular metabolism, and communication are associated with temporal and spatial ovule developmental asynchronies [40,88,98,99], incomplete penetrance of the character, and variable expressivity partially modulated by environmental conditions [100,101,102,103,104,105].

Even when these approaches allowed to identify several candidate genes and sequences associated with apomixis [64,73,74,80,90,106,107,108,109,110], most of them had no clear function assigned yet (however, see details in the next sections and under The Sexual Machinery and Apomixis-Like Phenotypes). Moreover, analyses of transcriptomes are complicated in most apomictic species by the high complexity, redundancy and polyploid nature of their genomes. The limitations of *de novo* transcriptomic assemblies in the absence of genomic information frequently lead to the compression of multiple gene copies into single transcriptome contigs, effectively reducing the gene space, but with potential loss of informative alleles [94,111].

## 4. Molecular Control of Apomixis in Apomictic Plants

Genetic and genomic studies were successful at identifying genes with a potential role in key steps of apomictic reproduction, especially genes involved in induction of fertilization-independent embryo development (for a detailed review, see [112,113,114]). Simultaneously, efforts for functional characterization of apomixis related genes and regulatory sequences had revealed an interesting diversity of proteins and small RNAs and long non-coding RNAs most likely being part of the apomixis regulatory cascade, yet their overall role and interconnection needs further evaluation. 

Among these findings, particularly diverse are those in *Paspalum*, such as the PnTgs1-like protein (a trimethylguanosine synthase-like protein), whose function has been associated gametophyte and possibly embryo development [115,116], *QUI-GON JINN* (a gene showing homology with mitogen-activated protein kinase kinase kinases (MAP3K/MAPKKK/MEKK)) having probable roles in the acquisition of a gametophytic cell fate by AIs, and the development of aposporous embryo sacs [110], *PsORC3a* (a pseudogene with homology to subunit 3 of the ORIGIN RECOGNITION COMPLEX (ORC3)) with a probable role regulating expression of its functional homolog and with the development of apomictic endosperm [64], *PN_LNC_N13* (a member of a family of long non-coding RNAs (lncRNAs)) involved in splicing regulation [117], or a number of small RNAs differentially represented in sexual and apomictic flowers likely involved in diverse regulatory pathways—including auxin signaling—important in promoting apomixis [118].

In *Boechera*, Corral et al. [108] and Mau et al. [109] functionally characterized genes associated with apomixis and formation of unreduced pollen (see details below). In addition, using laser-assisted microdissection to analyze transcriptomes of MMC and AIC, Schmidt et al. [88] found significant enrichment of different molecular functions, and differential expression of several core cell cycle genes, meiotic genes, and genes involved in epigenetic pathways. More recently, F-box genes and E3 ligases were identified among differentially expressed genes as probable regulators important for germline development in sexual versus apomictic ovular tissues [89]. Amiyete et al. [119,120] exposed several microRNAs, including one targeting a Squamosa promoter binding protein like (SPL11) differentially up-regulated at the MMC stage of ovule development in apomictic genotypes and small RNAs with potential binding sites in exonic regions, indicating a probable role in post-transcriptional gene regulation.

In *Eragrostis*, tissue-specific expression differences were detected for genes *AGO104* and *DMT102*, the first one is relevant for cell fate specificity, which may be preventing entry into meiosis and promoting gametophytic development in the diplosporous ovules, the second one is required for cytosine methylation at CNG sites likely involved in maintenance functions, which might be promoting the establishment of gametophytic specificities (i.e., four-celled embryo sacs) observed in apomictic ovules [121]. Analysis of small RNA families identified two genes, a MADS-box transcription factor gene, and a transposon, specifically repressed in the sexual genotype [122]. 

In *Hieracium*, using dihaploid and tetraploid apomictic plants and deletion mutants for apomixis (LOA) and parthenogeneisis (LOP) loci, Rabiger et al. [94] identified ovary-expressed ARGONAUTE genes and differentially expressed genes enriched for processes involved in small RNA biogenesis and chromatin silencing in apomicts, plus a small number of putative differentially targeted genes within the mutants and potential candidates in the specification and initiation of aposporous initial (AI) cells. Diverse Arabinogalactan-proteins (AGPs) were identified in AI and FM cells, expressed during early sexual and aposporous gametophyte development, which suggest a role in communication and signal transduction events leading to megaspore death and AI cell differentiation and progression toward gametogenesis [123].

In *Hypericum*, a truncated gene ARI showing homology to *AtARIADNE7* (encoding a ring finger E3 ligase protein involved in regulatory processes and ubiquitin-mediated protein) degradation, might have a role on protein populations altering gametophyte development [48]. In transcriptomic analyses, Galla et al. [85] found evidence that differentiation of the AI cell may be related to the misregulation of RNA-directed DNA methylation (RdDM) pathways (through *MEE57*, *CMT3*, and *IND2*, genes involved in the maintenance of DNA methylation, and AGO9, a component of RNA silencing complexes), chromatin-remodeling proteins (through HPCHC1, a MORC-like CW-type Zinc finger protein), and hormonal homeostasis (through HPPIN8, an auxin efflux carrier involved in ovule intercellular auxin gradients).

Studies on different apomictic systems indicate that epigenetic changes play a relevant role modulating apomixis progression (e.g., [124]) and controlling parthenogenesis. The methylation state of the genomic region for apomixis in *Paspalum* spp. controls the activation/repression of parthenogenesis [125], while in *Boechera* spp., aberrant imprinting and locus-specific DNA methylation changes are responsible for the maternal activation of a MADS-box gene encoding homologs of the transcription factor PHERES1 (PHE1), which promote embryo growth and is paternally expressed in sexual *Arabidopsis thaliana* [126]. In *Hieracium*, a paternally-inherited *PHE1* gene was identified as transcriptionally active in both sexually and apomictic derived seeds, suggesting the imprinting system may be modified [83]. Thus far, several proteins acting as putative inducers of parthenogenesis had been identified using differential transcriptomics (e.g., in *Poa pratensis* the SERK (for SOMATIC EMBRYOGENESIS RECEPTOR-LIKE KINASE); in *Pennisetum squamulatum*, the ASGR-BBML (for apospory-specific genomic region-BABY BOOM-like), and in the *Boechera holboelli*, complex APOLLO (for apomixis linked locus).

Albertini et al. [75] found a group of genes involved in signaling and trafficking events during reproductive development in *Poa pratensis*. The authors were able to identify *Pp*SERK as putative candidates for the induction of apomixis. Since SERK encodes a leucine-rich repeat containing a receptor-like kinase highly expressed during somatic and sexual embryogenesis *in vitro* [127], it was suggested that *Pp*SERK may promote autonomous embryo development in apomictic species [107]. A recent screen for members of the SERK family in *Paspalum notatum* resulted in the identification of two paralogs, *PnSERK1* and *PnSERK2*, the latter displaying a strong differential spatial expression pattern in ovules of apomict and sexual genotypes [128].

Another gene involved in the induction of parthenogenesis has been recently identified in the *Pennisetum/Cenchrus* species aggregate. The *PsASGR-BABY BOOM-like* (*PsASGR-BBML*) gene is expressed in egg cells before fertilization and can induce ectopic parthenogenetic embryos [129,130]. *PsASGR-BBML-like* is a member of the BBM-like clade of APETALA 2 transcription factors, which are conservatively widespread in the plant kingdom and functionally diverse [131]. Similar genes encoding BBM or BBM-like proteins were found to induce ectopic embryo formation in sexual model systems such as *Brassica* and *Arabidopsis* [132]. A functional test of the *PsASGR-BBML* transgene showed it promotes parthenogenesis and the production of haploid offspring in transgenic sexual pearl millet, can induce haploid embryo development in maize and rice at variable rates, but failed to induce haploid seed development in *Arabidopsis thaliana* [129,130,133]. These studies represent a good progress toward understanding of parthenogenetic development. Moreover, the ASGR region and the *PsASGR-BBML* gene sequences are highly conserved across the Paniceae in *Brachiaria* and *Panicum* species having different chromosomal backgrounds, suggesting a relevant role for the parthenogenesis component of apomixis [59,134]. However, the variable penetrance of the trait between transgenic lines and siblings of the same line, the complexity of embryo development observed in rice lines with shifts to vivipary and absence of endosperm development [130] plus the lack of a reference genome of the apomictic species highly restrict the options to deeply characterize the *PsASGR-BBML* gene and the network of genes and/or protein interactions which may further promote parthenogenesis. This limits the chances of manipulating plants with enhanced transcriptional levels (see [20] in this issue).

In *Boechera*, Sharbel et al. [86,87] were able to identify and characterize an apomixis-related gene (APOLLO; [108]). The APOLLO gene encodes an Aspartate Glutamate Aspartate Aspartate histidine exonuclease whose transcripts are down-regulated in sexual ovules entering meiosis while being up-regulated in apomeiotic ovules at the same stage of development in plants of the genus *Boechera* [108]. The gene has apo- and sex-specific alleles, both highly polymorphic (13 apoalleles and 21 sexalleles characterized out of 18 genotypes). All apomictic *Boechera* spp. accessions evaluated had at least one apoallele and one sexallele, i.e., they are heterozygous for the APOLLO gene, while all sexual genotypes were homozygous for sexalleles. Compared to the sexallele consensus sequence, dominant apoalleles are characterized by a set of linked apomixis-specific polymorphisms, of which the most relevant is a 20- nucleotide polymorphism present in the 5´ untranslated region (UTR) that contains specific transcription factor-binding sites for a number of known regulatory factors (e.g., ARABIDOPSIS THALIANA HOMEOBOX PROTEIN5; [108]). The authors suggested that the expression of a deregulated apoallele could induce the cascade of events leading to asexual female gamete formation in apomictic *Boechera* plants. However, thus far, apoalleles have not been validated and specific targets of apoalleles have not been identified yet. The high number of putative alleles (the lack of a consensus apomictic allele), the variation for APOLLO copy number (i.e., CNV) observed among genotypes, the lack of a similar genetic background in flanking regions of both apo- and sexalleles analyzed in three BACs (for details see [108]), and the complex evolutionary history of *Boechera* spp. [135] hamper clarification of apomixis initiation and development in this group.

## 5. The Sexual Machinery and Apomixis-Like Phenotypes

Over the last years, substantial work has been done in flowering plants on regards of gain- and loss-of-function mutants in different model systems, particularly in Arabidopsis, Petunia, and rice. These mutants had led to the identification of genes involved in different molecular processes during sexual seed development, including primordium differentiation, ovule development, gametophyte differentiation, and reproductive cell identity [136,137,138,139]. Several of such genes are associated with key steps of meiosis, embryo, and endosperm development and had shown phenotypes resembling elements of apomixis development [139,140,141,142,143,144]. However, thus far, all attempts of *de novo* engineering apomixis by, e.g., overexpression of candidate genes, has failed and today not a single fully (displaying high expressivity) apomictic mutant has been recovered from sexual species. I will mention here some relevant genes as an overview of the complexity behind the emergence of apomixis, as similar apomixis-like phenotypes can be caused by diverse genes, and all these mutants represent independent molecular events which should be or are expected to be developmentally “coordinated” in a natural apomictic plant.

### 5.1. Apomeiosis Mutants

MMC competence is restricted to only one cell from the nucellus in angiosperms, which undergo meiosis and give rise to the female gametophyte. Several genes and small RNAs interact to control gamete cell specification and meiosis by restricting the number of meiotic precursors or by enabling the meiosis progression [138]. The homeodomain transcription factor *WUSCHEL* (*WUS*) is a key regulator of stem cell fate, essential for the formation of the integuments and in the specification of MMCs. In *Arabidopsis*, Zhao et al. [145] demonstrated that genes coding KIP-RELATED PROTEIN (KRP) (an inhibitor of cyclin-dependent kinase, CDK) act to restrict the inactivation of the Retinoblastoma homolog RBR1 by CDKA;1. Since RBR1 is a repressor of *WUS*, *krp* and *rbr1* mutants display supernumerary meiocytes and embryo sac-like structures. Similarly, ICK (another family of CDK inhibitors) function to restrict the formation of megaspore mother cells and functional megaspores to one per ovule [146]. Inactivation of all ICK/KRP genes produces ovules having supernumerary MMCs, FMs and embryo sacs.

Small-RNA pathways have a relevant role in the regulation of female germline specification. Su et al. [147] found that THO non-cell complex autonomously restricts the megaspore mother cell fate to a single cell by preventing ectopic MMC formation via trans-acting small interfering RNA (ta-siRNA). The TEX1 protein of the THO/TREX complex, present only in epidermal cells, restricted the expression of ARF3 to the medio domain of ovule primordia through the biogenesis of TAS3-derived ta-siRNA. *TAS* family genes are transcripts targeted for cleavage by different miRNAs and ARF3 (AUXIN RESPONSE FACTOR) family members regulate leaf polarity, floral stem cell maintenance, and lateral root growth [148]. Mutations in components of the THO/TREX complex led to ovules with multiple megaspore mother cells, as well as a TAS3 ta-siRNA-insensitive mutant.

In another example of non-cell autonomous restriction of MMC fate, Zhao et al. [149] demonstrated that the *Arabidopsis* cytochrome P450 gene *KLU* (expressed in inner integument) produces a mobile signal that recruits the ATP-dependent chromatin, remodeling complex SWR1 to *WRKY28* (a transcription factor) in ovule primordia and promote its expression by the deposition of specific histone variants. *WRKY28* is expressed in somatic cells surrounding the MMC and is required to inhibit the acquisition of MMC-like cell fate. 

Members of the ARGONAUTE (AGO) protein family interact with sRNAs to regulate transcriptional and posttranscriptional gene expression through RNA-directed DNA methylation (RdDM) or ta-siRNA pathways. In rice the *MEIOSIS ARRESTED AT LEPTOTENE1* (*MEL1*) (an ortholog of *AtAGO5*) and the *MULTIPLE SPOROCYTE* (*MSP1*) genes initiate the sporogenous development, which regulates the completion of meiosis and the repression of germ cell fate in somatic tissues in rice ovules [150,151]. Mutants of those genes show an increased number of sporocytes, and multiple disorganized, but occasionally viable, female gametophytes [152,153] resembling the formation of multiple embryo sacs from nucellar cells in aposporous grasses [33]. 

Another case of apomixis-like mutants comes from *Arabidopsis* AGO9, a protein present in the cytoplasm of cells from the epidermal layer and accumulates in the nucleus of the MMC in ovules. AGO9 specifically binds 24-nt sRNAs derived from transposable elements, and together with RNA-DEPENDENT RNA POLYMERASE 6 (RDR6) involved in the biogenesis of siRNAs, they restrict the formation of multiple MMC cells in the ovule [152]. The archesporial cell of *ago9* mutants can differentiate directly into a functional megaspore without undergoing meiosis and produces aposporous-like embryo sacs [152]. This suggests that the transition from a sporophytic to a reproductive fate in cells surrounding the germline is suppressed by factors like AGO9. Another ARGONAUTE protein (AGO104) found in maize is coded by the *Dominant non-reduction4* (*Dnr4*) gene. Mutants of this protein produce viable unreduced female gametophytes due to defects in chromatin condensation during meiosis that finally fails to segregate chromosomes and creates a diplosplory-like phenotype. AGO104 is functionally related to the *At*AGO9 and is involved in gene silencing via methylation [144]. Diplospory is the type of apomixis found in *Tripsacum dactyloides*, a maize relative [154]. Interestingly, the AGO104 locus is located in a region on chromosome 6 of maize that is syntenic to the *Tripsacum* apomixis locus [153].

Maize DNA methyltransferases (DMT) are involved in RNA directed DNA methylation (RdDM). The inactivation of two DMT102 and DMT103 results in phenotypes evocative of the *At*AGO9 mutant, producing unreduced gametes and multiple embryo sacs in the ovule. Comparative analysis between the sexual maize biotype, *dmt* mutants, and apomictic hybrids exposed similar chromatin states in archesporial tissues of both the *dmt102* and apomicts [144]. Using *Arabidopsis* intraspecific hybrids between ecotypes and *ago9* and *rdr6* mutants, Rodriguez-Leal et al. [155] showed that multiple loci may control cell specification at the onset of female meiosis, and that variations in transcriptional regulation and localization of AGO9 and abnormal gamete precursors in *rdr6* ovules are reminiscent of natural phenotypic variation during megasporogenesis. In agreement with studies on apomictic plants (see previous section), these results suggest that epigenetic regulations play a role in the development and differentiation between apomixis and sexuality, and many plants will have the ability to reproduce asexually by short-circuiting the appropriate signals [152,156,157,158]. 

Studies in *Arabidopsis* identified the DYAD/SWITCH1 (SWI1) gene required for megasporogenesis (and microsporogenesis). SWI1 is expressed in female and male meiotic cells and have a role in sister chromatid cohesion and centromere organization [159]. SWI1 mutants show defects in female meiosis progression that led to a switch from meiotic to mitotic division and the formation of unreduced gametes in very low frequency [159,160,161]. The disruption of a single gene resulted in the bypass of meiosis thus resembling diplospory (a key component of apomixis). Another case of mutants showing apomixis-like phenotypes is the triple mutant called *MiMe* (“mitosis instead meiosis”) [162]. In *MiMe* plants, the combination of three mutants (*osd1*: omits the second meiosis; *Atspo11*-1: eliminates recombination and pairing; and *Atrec8*: modifies chromatid segregation) replace meiosis by a mitosis, and hence, apomeiotic-like embryo sacs are produced together with diploid gametes genetically identical to the mother plant [162,163].

Several mutants involved in polarity and cell fate determination in the *Arabidopsis thaliana* female gametophyte had been described (including cell cycle regulators, components of the cytokinin signaling pathway or MYB transcription factors) [114,139] showing arrested development at different stages, uncellularized gametophytes and cell identity loss, phenotypes similar to those observed in mature ovules of some apomicts carrying multiple embryo sacs. However, female gametogenesis in apomicts do not generally show drastic structural alterations, except perhaps for the ploidy change in nuclei and the absence of antipodal cells in embryo sacs of grass species.

### 5.2. Parthenogenetic Mutants

In both animals and plants, the egg cell activation event during sexual reproduction is highly reliant on pulse-signals that increase intracellular calcium ions [164] and can trigger in vitro parthenogenetic embryo development in animals [165]. In plants, however, similar experiments were insufficient to activate parthenogenesis in absence of fertilization [166]. Still, in the *multicopy suppressor of ira 1* (*msi1*) mutant in *Arabidopsis* the egg cell displays the capacity to start developing a parthenogenetic embryo without fertilization, undergoing abortion at later stages of development [143,167]. The *MSI1* gene in *Hieracium* (*HMSI1*) seemingly triggers the initiation of autonomous seed development [168]. Other potential candidate genes are those found to trigger somatic embryogenesis when expressed ectopically. These genes, identified mainly in *Arabidopsis* include *BABY BOOM* (*BBM*) [132] and *WUSCHEL* (*WUS*) [169] (both discussed above), *LEAFY COLTYLEDON 1* (*LEC1*) [170] and *LEAFY COTYLEDON 2* (*LEC2*) [171]. 

In maize a frame-shift mutation in *MATRILINEAL* (*MTL*), a pollen-specific phospholipase exclusively localized to sperm cytoplasm, triggered an increase in the haploid induction rate ([172]; see also [173,174]), with similar results in Indica rice [175] and wheat [176]. Even though *MTL* is not directly involved in parthenogenesis, it shows that pollen-specific genes may mediate the formation of haploid seeds. Haploid induction is triggered by nuclear-cytoplasmic interactions in durum wheat [177] and other species, and is spontaneously (and rarely) observed in natural apomicts [21] highlighting that specific combination of non-nuclear sperm components may provide fertilization-like signals promoting parthenogenetic egg cell progression. 

In Bryophytes, the homeobox gene *BELL1* and *RKD* family of transcription factors might be of relevance for ectopic induction of embryogenesis in angiosperms. Overexpression of *BELL1* induces embryo formation in the moss *Physcomitrella patens* and reproductive diploid sporophytes from gametophytic cells without fertilization [178]. While *BELL1* is a central molecular trigger for the gametophyte-to-sporophyte transition in *P. patens*, in rice *BELL1*-like homeobox genes regulate inflorescence architecture [179] and in *A. thaliana BELL1* interacts MADS box factors and repress *WUS* for proper differentiation of the ovule [180]. In the liverwort *Marchantia polymorpha*, transformants using artificial microRNAs (amiRNA) constructs to downregulate endogenous *MpRKD* transcript levels show severe defects in gemma cup formation and egg-like cells underwent cell divisions in the absence of fertilization before the archegonia reached maturity [181]. However, the pattern of such divisions differed from those of the wild-type embryo, and failed to form viable, normally patterned embryos [181]. Thus, *MpRKD* has a crucial role in the formation of the gemma cup and in establishing and/or maintaining the quiescent state of the egg cell prior to fertilization, preventing it from undergoing parthenogenetic-like grow by cell divisions. Whereas *MpRKD* appears to be primarily active in the control of gametophyte development, in flowering plants RKD function is recruited to early phases of embryogenesis but loss-of-function experiments in *A. thaliana* have not yet produced phenotypes affecting the egg cell [182].

Differences observed among plants in haploid and diploid parthenogenesis likely underly a combination of genetic redundancy and divergent evolution in regulatory factors ensuring egg cell quiescence prior fertilization [183].

### 5.3. Mutants Associated with Endosperm Development

Despite the biological and economic relevance of the endosperm, the molecular events underlying its development were largely unexplored. In recent years, this has changed with the combined use of mutants, cell specific markers, and plant hormone sensing reporters [184]. Additional loss-of-function *Arabidopsis* mutants associated with apomixis-like phenotypes are those of the Polycomb group complex 2 (PRC2) that repress endosperm formation and seed development in the absence of fertilization. The genes of this complex encode several proteins, such as MEDEA (MEA) [185], FERTILIZATION INDEPENDENT SEED 2 (FIS2) [142,186], FERTILIZATION INDEPENDENT ENDOSPERM (FIE) [141], and the mentioned MULTICOPY SUPPRESSOR OF IRA 1 (MSI1) [143]. *FIS* class genes are expressed in the central cell of the female gametophyte and encode protein subunits of the PRC2 that control the expression of downstream genes promoting endosperm growth (e.g., *MADS-box PHERES1* or *PHE1*) by silencing paternal or maternal alleles [88,187,188,189]. FIS-PRC2 mutants lead to mitotic activation of the central cell nucleus division in absence of fertilization and the autonomous development of a diploid endosperm [189]. However, like in the majority of mutant phenotypes, in most *fie* seeds the endosperm can only growth until the cellularized phase and the development of the embryo is arrested at globular-heart stages [141,167,190]. Unexpectedly, specific downregulation of *FIE* in sexual *Hieracium* failed to stimulate the autonomous proliferation of the central cell, and pollination was required to activate seed initiation but led to subsequent seed abortion [191]. Similarly, in *FIE* downregulated apomictic *Hieracium*, autonomous embryo and endosperm initiation were also inhibited and seed development was arrested, with most autonomous seeds displaying defective embryo and endosperm growths [191]. Thus, FIE might not be directly involved in endosperm progression in apomicts (and hence is likely to be a poor candidate for the synthesis of apomixis in crop plants [191]); alternatively, it might require interactions with other factors and/or molecular signals to stimulate endosperm progression without fertilization.

## 6. Overall Features, Hypotheses in Conflict, and the Quest for the Origin of Apomixis

The characterization of a number of genes with likely relevant roles in the activation and/or progression of apomeiosis and/or parthenogenesis in different plant groups and the observed potential of many taxa to express apomixis-like traits indicates that the induction of apomeiosis and/or parthenogenesis by genetic and epigenetic manipulation is feasible. However, the evidence also indicates that apomixis is caused by a cluster of genes—physically linked or not—most likely functionally related that act coordinately in the activation of apomixis. Thus, while the use of different genetic modification techniques (e.g., Transgene-directed Mutagenesis, RNA-directed DNA-Methylation, or new restriction enzyme techniques (gene-editing); [192]) in attempting to induce apomixis in a sexual plant is likely the best strategy, expectations about obtaining a neoapomict may not be realistic. The association of apomictic sequences to accumulation of transposable elements, gene degeneration, and gene-specific silencing mechanisms (likely based on chromatin remodeling factors or transacting and heterochromatic interfering RNAs involved in both transcriptional and post-transcriptional gene regulation, see [139,193,194]) suggests that inducing apomixis *sensu stricto* by simply knocking out meiotic genes is unfeasible. A synthetic approach to engineering apomixis by targeting key regulatory steps has been considered for a long time while many of the above discussed genes were discovered [195], and yet, achievements on this goal progressed slowly (for details see [113]). Examples are the feeble penetrance attained by synthetic apomixis-like mutants (see for example [162]), which expose the fact that changing radically key developmental steps likely require the coordination of multiple metabolic pathways, molecular signals, and cellular players. Unless a proper understanding of the genetic architecture of apomixis exists, the prospect for a genetic engineered apomictic crop remains equivocal (but see [20] other reviews in this issue).

Thus far, three main hypotheses (HA-C) are usually referred to for the cause of apomixis when trying to explain the varied facts and features around the formation of clonal seeds:

H**A**) it is caused by developmental asynchronies (de-regulation of the sexual developmental pathway) resulting as a consequence of hybridization and/or polyploidization [9,195,196,197]; this hypothesis is supported by ovule development analyses and transcriptional profiling (see discussion above).

H**B**) it is a mutation-based anomaly that involves a simple or (conceivably) complex genetic locus [15,198]; this hypothesis is supported by natural and induced mutants (see discussion above).

H**C**) it is an ancient state from which sexual reproduction arose as an alternative, epigenetically regulated and with remnant capacities relatively conserved across eukaryotes [199,200], supported by mutants affecting methylation pathways (see discussion above).

Functional apomixis needs the coordinated concurrence of three reproductive elements (i.e., apomeiosis or the formation of unreduced non-recombinant gametes, embryogenesis without fertilization, and endosperm development), each with a complex developmental background in sexual plants (e.g., involving fundamental changes in cell specification and fate during ovule and seed developments [201,202]) and several genes involved (see above). Even though these hypotheses are not mutually exclusive, they fail to provide an appropriate frame and a unified explanation covering all empirical data about apomixis. While the simultaneous deregulation in multiple loci (H**B**) can be a downstream effect initiated by mutations on genes controlling apomeiosis, parthenogenesis, or endosperm development (H**A**), it is highly unlikely that these three changes on the sexual reproductive program could evolve together in a sexual ancestor through randomly occurring mutations. As any single step has a negative effect on the fitness of its carrier [11,203,204], the needed genetic changes (and basis) for apomixis must appear concurrently and have a concerted expression to be functional and produce a viable seed (details about the first stages of emergence and stabilization of apomixis in natural populations can be found in Hojsgaard [205] and Hojsgaard and Hoerandl [30]). Likewise, as apomixis can appear in the F_1_ progeny from sexual parents after hybridization and polyploidization [99,206], the emergence of apomixis may not require long-term evolutionary changes or separate evolution of mutants for each reproductive step involved in apomixis. This is a relevant point when considering apomixis breeding. Similarly, apomixis is not simply a consequence of polyploidization since 1) most known polyploid plants are sexual; 2) polyploid apomictic species do not exhibit recombinant sibs able to reproduce only by sexuality in nature (as expected if polyploidization would separately affect genes related with apomeiosis, parthenogenesis, and endosperm formation); and 3) polyploid sexuals can be recovered by inducing chromosome doubling in sexual diploids from agamic complexes [207]. Thus, the emergence of apomixis in a plant is likely based on a specific molecular mechanism that has cytological consequences and not *vice versa*. In sexual-apomictic comparisons, sexuals display a normal transcriptional profile ascribed to (stable) meiosis + syngamy (in other words, non-functional apomixis) and apomicts display a general de-regulation of genes and several sequence-level changes, ascribed to the abnormal meiosis and irregular syngamy usually accompanied by high rates of apomeiosis + parthenogenesis (or functional apomixis) [38,40]. 

This “all-or-nothing” pattern observed at developmental and gene expression levels in different apomicts—together with the other mentioned features—implies that apomixis in plants is triggered by a (likely common) single-event molecular mechanism that must implicate simultaneous changes on genes “masterly” associated with (at least three) critical steps of the sexual program (H**A** and H**C**), and involve a corresponding de-regulation of many more loci (H**B**) leading to the reproductive switch (and associated shift of ploidy) that deplete sex (meiosis and syngamy) and activates apomixis (apomeiosis and parthenogenesis) during seed formation.

Only by unveiling the molecular nature of such allegedly single event that prelude the emergence of apomixis in natural plant species will we be able to have a chance to effectively manipulate genetic sequences and sexuality towards a penetrant apomixis-like development in plants.

## 7. Current View and Prospects

Disentangling the molecular basis and regulatory mechanisms underlying the emergence of apomixis requires species-specific knowledge about the (epi)genetic and genomic contexts. For doing so, in general terms, it is necessary to compare apomictic genomes against sexual genomes, preferentially of the same ploidy and with no hybrid origin to enable the identification of all changes in gene sequences and genome structure required to activate apomixis. Knowledge about the genomic background interacting with an apomixis locus will provide indirect but valuable information about (primary and secondary) variants of metabolic relevance and interrelation of regulatory control mechanisms in the expression of apomixis (like cell cycle control, protein turnover, signal transduction, and hormonal regulatory pathways). Mainly due to the cost and complexity of such projects (almost all apomicts are polyploids and/or hybrids), no genome of an apomictic species has been yet sequenced. The publication of the perl millet genome [*Cenchrus americanus* (L.) Morrone] (genome size ~1.79 Gb; [208]), a minor crop and diploid sexual relative of tetraploid apomictic *C. ciliaris* L. (Buffelgrass), as well as of the weeping lovegrass genome [*Eragrostis curvula* (Schrad.) Nees] (genome size ~660 Mb; [209]), a diploid sexual forage grass having apomictic polyploid cytotypes, and the genomes of *Boechera stricta* (genome size 216 Mb) and *B. retrofracta* (genome size 200 Mb) [210,211,212], two sexual diploid herbs relative to apomictic *B. holboellii* and *B. divaricarpa*, represent an effort in that direction. Two recently-initiated projects aiming at sequencing the genome of several sexual and apomictic *Boechera* species in Canada [213] and Switzerland [214], and the generation of an apomictic *Hieracium* (predicted genome size 3.6 Gb) genomic and transcriptomic resource [94], may provide the first datasets for sexual-apomictic pair genomic comparisons. 

However, the NGS market has rapidly evolved and a large number of developments have gained accuracy and speed, have partially solved technically complex issues (e.g., genome phasing), and reduced the cost of sequencing [215,216,217]. Hence, it is timely to progress the state-of-the-art and join efforts in sequencing high quality genomes using long sequence reads to cover repetitive regions (present in some apomixis loci) and linked-read strategies to improve genome contiguity and assembly quality. Different cost-effective and scalable strategies are offered in the market. Convergence between assembled sequences and chromosome number can be achieved through different chromosome conformation capture techniques [218] and scaffolding strategies enhancing sequence continuity and the integration of fragmentary sequences; however, it needs to be decided on a case-by-case basis. Even when genome phasing might not be strictly necessary for the identification of apomixis genes (the simplex haplotype of an apomixis locus is expected to be sufficiently divergent to other sexual haplotypes), defining the most likely phase within a set of connected variants of an allele and appropriate read population phasing [219,220] will improve the ability to correctly identify compound heterozygosity at rare variants in either *trans* or *cis* interactions relevant for apomixis expressivity in both wild and mutant plants. The causality for the observed variation in penetrance and expressivity among natural apomicts and apomixis-like mutants is widely undetermined. Integration of genetic maps and methylomes, when available, into quality genomes will certainly facilitate linking phenotype variability to genotypes and identifying genetic modifiers. Despite that complex, quantitative phenotypes are stably inherited across generations [8], phenotypic instability, likely due to gene-environment interactions or genetic compensation to transgenes, is known for natural apomicts and mutants [8,181]. Hence, combined evaluations of transgenerational phenotype stability and genomic data on apomictic plants and apomixis-like mutants, including possible off-target effects in genome editing, will be required to unclose molecular interactions toward penetrant apomixis phenotypes. 

Yet, apomixis research is underfunded, and not every newly available technique is a necessary improvement for solving the molecular control of apomixis in each plant system. Comparative studies (such as coexpression or GWA studies) can help identifying shared elements in expression networks and coordinated activity of gene sets in complex phenotypes [81]. However, methodological differences often neglect integration of multiple data sets from different sources due to incompatibility. Perhaps, given the relevance and remaining challenges toward apomixis technology, thinking of an international consortium between public and private institutions enabling synergies between qualified researchers, know-how and access to funding to implement apomixis technology in main crops is not utopian. By stating clear rules and goals among participants, such an enterprise will facilitate financing and speed up the collection of data, analysis, and results validation while reducing individual efforts and maximizing data compatibility. 

Even though the endeavor to get an apomictic crop is still technically demanding and challenging (e.g., underlying gene networks and regulatory pathways differ among apomicts, and hence, manipulating those to shift developmental pathways will present distinct hurdles [144,183]), it will help us realize whether implementing apomixis breeding in agroscience is a feasible task or not, a pending topic that is becoming critical and will help humanity resolving current and future needs on food supply, landscape management, and conservation of biodiversity areas.

## Figures and Tables

**Figure 1 genes-11-00411-f001:**
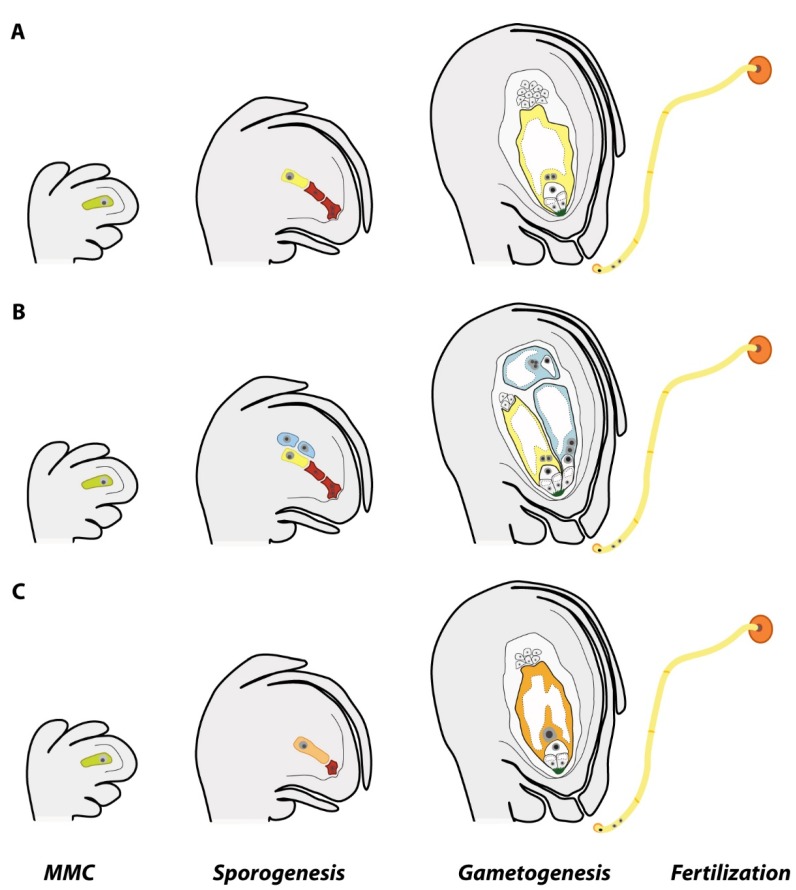
Reproductive alternatives in ovules of angiosperms with gametophytic apomixis. (**A**) Standard events during sexual reproduction, leading to the specification of a megaspore mother cell (MMC), who goes into meiosis and forms a row of three–four megaspores (end of sporogenesis), while the chalazal one (in yellow) develops further into a meiotic female gametophyte carrying haploid nuclei (end of gametogenesis). During fertilization, two haploid sperms fuses to the egg and central cells each to form the embryo and the endosperm of the seed, respectively. (**B**) In aposporous species, simultaneously with the MMC meiotic division, aposporous initial cells (in light blue) appear in the nucellus and acquire a megaspore-like fate entering gametogenesis and developing aposporous (diploid) female gametophytes. During fertilization, one haploid sperm fuse to the central cell to form the endosperm and the egg cell develops parthenogenetically into an embryo. (**C**) In diplosporous species, the MMC goes through a modified meiosis and form a diploid megaspore (in orange) and later a female gametophyte. During fertilization, one sperm fuse to the central cell and develop the endosperm and the embryo is developed by parthenogenesis. The drawing is based on observations in *Paspalum* spp.

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
