# Peer review of "Apomixis Technology: Separating the Wheat from the Chaff"

_genes, 2020, doi:10.3390/genes11040411_

Round 1
Reviewer 1 Report
Apomixis, a natural way of clonal reproduction through seeds found in many plant species but not crops, has been long considered as a promising tool for plant breeding and agriculture but genetic and molecular studies in the last three decades have shed little light on its genetics and molecular control. In this Opinion report, the author intends to present approaches and strategies for the delivery of the apomixis technology to plant breeders and farmers. Although the topic is of general interest, the manuscript suffers from several weaknesses listed below and that, in my opinion, need to be addressed prior to publication in Genes.
1- As the manuscript deals with apomixis technology delivery, it should not be restricted to the understanding and mimicking of natural systems. In my opinion, one section should be dedicated to de novo engineering of apomixis in plants. The MiMe mutant, first obtained in Arabidopsis, is a strategy for installing functional apomeiosis that is likely to work in crop species as shown for rice. On the same line, strategies for inducing maternal embryos are also available (i.e. tail swap approach) and were capable of producing clonal progenies when combined with MiMe mutants. Propects and limitation of theses efforts could be also discussed.
2- The Introduction section is about 25% of the manuscript length and it needs to be significantly reduced. There is no need to detail the reasons why apomixis is a promising and awaited tool for plant breeding and agriculture considering recent reviews on the topic.
3- Since Genes scope covers genetics and genomics at large, some parts of the manuscript (section 2 in particular) sound difficult to understand for readers that are not familiar with plant sciences and, particularly, plant reproductive biology. A short paragraph describing sexual plant reproduction (possibly including a simple drawing if possible) would certainly help non specialist readers to understand how apomictic phenotypes are built (or expected to be) from the sexual pathway.
4- Section 4, that deals with the molecular control of apomixis as understood from studies in apomictic systems, lacks references to regulatory mechanisms (e.g. sRNA and Lnc-RNA; alternative splicing; cell cycle regulation) and possible genes involved reported in several hallmark apomictic species (e.g. Hypericum: Galla et al., 2017; Citrus: Long et al, 2016; Hieracium: Rabiger et al., 2016; Boechera : Amiteye et al., 2011 and 2013, Zühl et al 2019; Paspalum : Siena et al, 2014, Ochogavia et al. 2018, Ortiz et al, 2019, Colono et al. 2020; Eragrostis : Selva et al;, 2017 and Garbus et al, 2019).
5- Section 5 that deals with works in sexual species also lacks important references, particularly for reproductive cell identity i.e. repressors of the number of MMCs within the ovule (eg KRP/CDAKA/RBR1; ta-siARF; AGO9/RDR6/SG3; KLU/SWR1/WRKY28; THO complex/TAS3-ARF3), gametophyte differentiation (eg AGO5; MYB64), egg cell differentiation (eg PsAGSR-BBML; PpBELL1; MpRKD) (Su et al., 2017; Zhao et al., 2017; Nakajima, 2018; Hisanaga et al, 2019,Tucker et al., 2012) and haploid formation (e.g. Liu et al. 2017; Kelliher et al. 2017; Gilles et al. 2017).
6- The last section posits the main effort on genomics to resolve the molecular basis of apomixis and ultimately establish apomixis in sexual crops. Although I fully agree on this, other areas appear critical for validating candidates and delivering the apomixis technology, i.e. monitoring and altering gene expression in reproductive tissues, transgenerational impact of asexual reproduction on phenotypes.
7- References. The number of references is quite high (175 in total) and should be reduced significantly. Some recent/important reviews of interest are missing (e.g. Leon-Martinez and Vielle-Calzada, 2019; Grimanelli et al. 2001 regarding apomixis as a phenotype resulting from heterochronicity). Also the list needs extensive revision, i.e. mis-citation (#40 p. 3 for facultative apomixis), use more recent reviews (i.e. #63 from 1990 on the evolution of sexual chromosomes).
Author Response
Specific responses to the main points raised by Reviewer 1:
1- I reckon the relevance of approaches attempting de novo engineering apomixis into sexual plants. The reason why I haven´t dedicated a section to such approaches and I only mentioned them briefly (but enough for the point made on each section) is mainly because those approaches will be presented in detail in other contributions planned within the same Special Issue, specifically addressing prospects and limitations of GE approaches. Since here the main point of the manuscript is to stress the fact that still little is known about the molecular control of natural apomixis (in apomictic plants) and how to potentially deliver solutions to it, there is no need to present (or repeat) in this manuscript more specific aspects of de novo apomixis in sexual plants.
2- I shortened the introduction and now it is 847 words long (847/7207= 11.75%).
I recognise that reasons about the relevance of apomixis in plant breeding probably overlap other manuscripts, this involve now only the first two paragraphs and are necessary to make a point (why plant breeding approaches need apomixis technology), take the reader into the problem (many attempts have so far failed), and place the direction of the current manuscript.
I hope the new relative size of the introductory section meet the reviewer’s expectations.
3- I agree. I have added a figure (Figure 1, page 3) showing the main aspect of the sexual pathway and observed deviations in the main types of gametophytic apomixis.
4- Thank you for this observation. Since this contribution is not intended to be a review per se, I first tried to focus only in relevant genes with some functional characterization. Now, I have added a specific paragraph mentioning these other studies (lines 199-250) plus sentences including the pertinent literature along the text (lines 158-161; line 177; line 252).
5- Yes, it lacked those references. I know. Due to space and because of the complexity of meiosis, I have specifically written in the first paragraph that I would mention only some of all genes to some extend related to reproductive steps in sexual shared with apomicts.
Now, I added the new suggested references and a few paragraphs including more specific information on this topic (lines 327-356, 361-365, 381-385, 400-407 and 420-447).
6- I agree. I understand monitoring and altering gene expression is a direct or indirect part of most described papers on characterization of genes and mutants. The transgenerational impact of apomixis on phenotypes has been already addressed by Sailer et al. 2016 (Curr. Biol. 26) and cited in the text.
I now added a paragraph considering some of these aspects on lines 576-607.
7- Thank you. I agree, the reference list is long. I replaced some references and added the missing reviews (plus the new references from above points 4-5). Reducing more the reference list will be somehow difficult, most citations are used twice or more and in very specific contexts. Since there is no restriction on manuscript length and number of references, it would be appropriate keeping original references.
I have changed reference 40 (line 142) for Hand et al. 2015 (Heredity 114) and reference 63 (line 170) for Charlesworth 2017 (Phil. Trans. R. Soc. B 372).
Reviewer 2 Report
Dear Editor,
The review presented by Hojsgaard, describes the more recent achievements on the research on apomixis in the perspective to develop an apomixis system to be introgressed in crops by appropriate methods. The manuscripts is well written and the questions raised in the introduction are well addressed. Of particular interest, to my opinion is the discussion of the architecture of the apomixes locus in relation to evolutionary pathways of natural apomictic species. In this perspective what clearly emerges is the necessity of studying the molecular mechanisms of natural apomictics as preparatory for the developing a stable apomixis system to be introgressed into crops. In fact, all mutations that concurred at the formation of natural apomicits have been the chance to be stabilized or discarded by natural and therefore only such machinery should be exported to crops to obtain an apomictic crop that is stable over time. For these reasons, I believe the manuscript be of interest for a wide range of readers including basic reproductive biologists and plant breeders, and on the basis of these considerations I strongly support publication of this manuscript as it stands proved the Author is ready to include the minor suggestions below,
Minor remarks:
- Developmental features of apomixis
-) Lines 110-116. The fact that differently from diplospory, meiosis, in aposporous apomictic systems is neither bypassed nor circumvented, but rather it parallels aposporous embryo formation until the formation of functional macrospore is not sufficiently highlighted. This is important because aposporous development affects cell line that is different from that of diplospory.
-) Lines 125-126. The condition of 2m:0p of genome contribution in the endosperm is enveloped as a particular case of pseudogamous apomixis. In my opinion if would be more formally correct to describe this condition as full autonomous apomixis.
- Molecular control of apomixis in apomictic plants.
-) Lines 204. I guess the correct name is “APETALA”
-) Lines 201-214. When mentioning the BBM-like gene, not sufficient emphasis is given to the fact that this is the only candidate that shows a clear apomixis-related phenotype and it is genetically linked to apomixis. Although this fact is implicit on the acronym ASGR, this might be unknown by readers not expert in apomixis.
- The sexual machinery and apomixes-like phenotype
-) Line 248. Have failed
-) Lines 299-300. When mentioning “autonomous seed development” please specify if possible if this is related to embryo, endosperm or both.
Author Response
Specific comments about minor remarks from reviewer 2:
2-Developmental features of apomixis
-) Lines 110-116
Thank you for this observation. I reckon that in aposporous plants meiosis is still functional (not in all ovules) and may develop into a reduced embryo sac (only or mainly in grasses). However, this is a relevant issue in an evolutionary frame in which the possibility of having recombinant progeny can affect gene evolutionary trajectories. In the case of this paper, as the relevant point relies on the practical consequence of apomixis in which residual levels of sexuality are not important.
Anyhow, I added “independent of the MMC progression” (line 124) in the suggested sentence to make clear this point. I would also like to stress that the use of “avoidance” and “circumvention” (as synonym for avoidance) in the original text were aimed at making clear that meiosis is not always “bypassed” as in diplospory but rather skipped because somatic cells are not committed to go into meiosis. I have now changed “avoided” by “prevented or skipped” (line 119) to stress such difference.
-) Lines 125-126.
I agree. In fact, it is clarified between brackets in the text “(from 2m:0p in embryo sacs with autonomous endosperm development, till 1m:1p till 8m:1p in pseudogamous embryo sacs)” (lines 132-133).
4-Molecular control of apomixis in apomictic plants.
-) Lines 204.
That is right, thank you. The typo has been now corrected (line 275).
-) Lines 201-214.
I added now the meaning for ASGR (“for apospory-specific genomic region”; line 262) in the first paragraph of this section and uniformized presentation of abbreviations for all three cases. I should also mention that the section title is “Molecular control of apomixis in apomictic plants” which make clear that the genes discussed here are described from apomictic plants and no sexual ones.
The sexual machinery and apomixes-like phenotype
-) Line 248.
Thanks. Corrected (line 319).
-) Lines 299-300.
Line 415. Since the seed is only conceived as a combination of both embryo and endosperm tissues, it is possible to assume that both are related to the autonomous development. In the case of apomictic plants, all embryo develops autonomously (i.e. by parthenogenesis) even if activation of the endosperm is required. In the case of the gene HMSI1, its function is affecting the autonomous development of the seed. It is discussed in lines 461-473.
Round 2
Reviewer 1 Report
I would like to thank the author for the clarifications and the additional figure. The manuscript has been substantially improved and, therefore, I can accept to endorse its publication in its current form.
However, I have one concern with the number of references which is over 200 in the new version (175 in the original ms, currently 220). In my opinion, and despite the Genes policy of not restricting manuscripts length I fully understand and support, this is far too high for an opinion paper relatively short in length and will not help new readers to become more familiar with the challenges in apomixis research and its use in breeding and farming. As I said in my first report, I would have prefered the author to reduce the reference list significantly. This can be achieved by using more often reviews, eg Grimanelli (Curr Op Plant Biol 2012) for pioneer works in epigenetics and apomixis, Ozias-Akin and Conner (TIG, 2019) for BBMLike/BBM related works, Ortiz et al. (Ann Bot 2013) for many of the work in Paspalum,...